# Usefulness of Orientation to the Year as an Aid to Case Finding of Mild Cognitive Impairment or Depression in Community-Dwelling Older Adults

**DOI:** 10.3390/ijerph18158096

**Published:** 2021-07-30

**Authors:** Hanhee Bae, Sunyoung Kim, Byungsung Kim, Miji Kim, Jisoo Yang, Eunjin Jeong, Yoonki Kim, Changwon Won

**Affiliations:** 1Department of Family Medicine, Kyung Hee University Medical Center, Seoul 02447, Korea; hanhee0703@gmail.com (H.B.); k23123@naver.com (Y.K.); 2Department of Family Medicine, College of Medicine, Kyung Hee University, Seoul 02447, Korea; ggutsun@naver.com (S.K.); byungskim@naver.com (B.K.); wltn-0208@hanmail.net (J.Y.); ejjeong312@gmail.com (E.J.); 3East-West Medical Research Institute, College of Medicine, Kyung Hee University, Seoul 02447, Korea; mijiak@khu.ac.kr; 4Elderly Frailty Research Center, Department of Family Medicine, College of Medicine, Kyung Hee University, Seoul 02447, Korea

**Keywords:** cognitive dysfunction, depression, orientation

## Abstract

Mild cognitive impairment (MCI) and depression are common and frequently misdiagnosed in older adults in primary care. In particular, depression combined with cognitive dysfunction is associated with a higher risk of dementia. We tried to find the usefulness of orientation to time as an easy case-finding tool for suspecting MCI or depression. This cross-sectional study included 2668 community-dwelling adults aged 70–84 years from the Korean Frailty and Aging Cohort Study (mean age of 76.0 ± 3.9 years). MCI was defined based on the criteria from the National Institute on Aging and the Alzheimer’s Association; depression was defined as a score of ≥6 on the Geriatric Depression Scale—Short Form (GDS-SF). Time orientation to year, month, day of the week, date, and season were tested. The sensitivity for the diagnosis of each of MCI and depression was the highest for the orientation to year (MCI, 17.7%; depression, 16.0%). For the diagnosis of MCI or depression, orientation to the year had the highest sensitivity (15.5%), and the specificity, PPV, NPV was 95.5%, 67.0%, 65.5%. In conclusion, asking “what year is it?” can be helpful as an aid to case finding to suspect MCI or depression in community and primary care settings.

## 1. Introduction

Rapid global aging and subsequent life expectancy prolongation have raised many medical concerns. With increasing life expectancy, cognitive impairment is being recognized as major threats to healthy aging and quality of life [1]. In the U.S., an estimated 5.8 million people aged 65 years and older were living with dementia in 2020 [2]. In South Korea, the number of patients with dementia has been increasing and is expected to reach nearly 3 million by 2050 [3]. In addition, the dementia prodrome, mild cognitive impairment (MCI), has garnered attention and active study since the early 1990s as a risk factor for developing dementia, with an annual rate of progression of 12%, reaching up to 20% in populations with prodromal Alzheimer’s disease or stroke [4,5]. In primary care settings, the diagnosis of dementia can be delayed by up to 2.5 years [6]. Therefore, early screening and diagnosis of MCI are important, as early detection of cognitive decline can allow interventions before further damage occurs and may postpone or even prevent progression to dementia [7].

There are numerous cognitive screening tools with several limitations, including low accuracy in diagnosing MCI [8]. Out of the many screening tools for dementia, the Mini-Mental State Examination (MMSE) is the measure most widely used to screen for cognitive dysfunction [9]. Of the items examined in the MMSE, orientation in time and delayed recall are known to be more closely related to Alzheimer’s disease [10]. In particular, those with at least one error in the time-orientation domain are known to have a 58% higher risk of dementia [10]. Nevertheless, few studies have examined the links between time-orientation errors and MCI.

Depression is a common problem in older adults. In community settings, approximately 5% of older adults aged ≥65 years meet the diagnostic criteria for major depression, and 8%–16% meet the criteria for minor depression [11]. Depression in old age is related to several diseases and worsening of health. Depression increases the risk of falls in older people by almost 50% [12]. A major problem is that depression is often undetected and untreated [13]. According to one meta-analysis, primary care physicians have correctly detected only 47.3% of patients with depression among older adults over 60 years of age [14].

The prevalence of depression among demented patients has been reported to be up to 60% [15]. In addition, it is well known that depression combined with cognitive dysfunction is associated with a higher risk of cognitive impairment compared to cognitive dysfunction alone [16]. Therefore, an easy case-finding tool for suspecting depression or MCI in older adults in primary care is needed.

Based on this context, we examined the value of the time-orientation test as an aid to case finding for MCI and/or depression.

## 2. Materials and Methods

### 2.1. Participants

Adults who participated in the Korean Frailty and Aging Cohort Study (KFACS) were aged between 70 and 84 years. The KFACS is a multicenter, longitudinal study with a baseline survey conducted in 2016–2017 [17]. A total of 3014 community-dwelling older adults from urban and rural regions nationwide in 10 study centers were recruited after stratification based on age and sex [17]. Each center recruited participants using quota sampling stratified by age (70–74, 75–79, and 80–84 years with a ratio of 6:5:4, respectively) and sex (male and female), with the aim of recruiting 1500 men and 1500 women, ending up oversampling with the disproportion of sex. Finally, 3014 participants were enrolled with 1432 men and 1582 women. Participants were recruited from diverse settings (local senior welfare centers, community health centers, apartments, housing complexes, and outpatient clinics) to minimize selection bias [17].

Participants with a history of dementia were excluded from the study. The collected baseline data were demographic, medical, psychosocial, socioecological, and cognitive functions. Among the participants, only 2668 who completed questionnaires required for the diagnosis of MCI were included in this study.

### 2.2. MCI

The National Institute on Aging and the Alzheimer’s Association revised the criteria for MCI in 2011 [18]. The revised criteria for MCI are as follows: (1) cognitive concern reflecting a change in cognition reported by the patient, informant, or clinicians (i.e., historical or observed evidence of decline over time); (2) objective evidence of impairment in one or more cognitive domains, typically including memory (i.e., formal or bedside testing to establish the level of cognitive function in multiple domains); (3) preservation of independence in functional abilities; and (4) absence of dementia.

Using this, we defined MCI as follows:

(1) Cognitive function test results <−1.5 standard deviations of those of age-, sex-, and education-matched Korean norms on any one of the four cognitive functioning tests including the trail-making test, frontal assessment battery, digit span backward, and word list recall test [1].

(2) No significant dependency on instrumental activities of daily living (IADL) assessed using three items of the K-IADL: managing money, telephone use, and responsibility for medication [1,19].

We regarded participants who fulfilled criteria (1) but had significant dependency in IADL as having dementia; therefore, they were excluded from this study.

### 2.3. Depression

In this study, we regarded depression as a score of ≥6 on the Geriatric Depression Scale—Short Form (GDS-SF), which is a screening measure designed for detecting depression in older populations. All participants with a score of <6 were grouped in the non-depression group. The GDS-SF consists of 15 items [20,21].

### 2.4. Neuropsychological Tests

#### 2.4.1. The Consortium to Establish a Registry for Alzheimer’s Disease Assessment Battery (CERAD-K)

We conducted neuropsychological tests, the Consortium to Establish a Registry for Alzheimer’s Disease Assessment Battery (CERAD-K) and Korean version of the Frontal Assessment Battery, to appraise comprehensive cognitive function. The CERAD-K is a standardized clinical and neuropsychological assessment battery for the evaluation of patients with Alzheimer’s disease. The CERAD-K consists of eight tests (verbal fluency, modified Boston naming, MMSE, word list memory, constructional praxis, word list recall, word list recognition, and constructional praxis recall); however, in this study, word list memory/recall/recognition, digit span (forward and backward), the trail-making test (TMT) A, and the MMSE were included [22].

The Word List Recall: evaluates the ability to recall the given 10 words from the word list memory task. A maximum of 90 sec is allowed, and the maximum score is 10 [22];Digit Span Backward: consists of 2 to 8 digits. The participants are given digits at a rate of one digit per sec and had to repeat numbers in the reverse order. The score is the total number of correct items [22];The Trail-Making Test: evaluates attention, ordering, executive function, time-space search, and mental motion velocity. The participants were asked to draw a line connecting the numbers from 1 to 25 in ascending order, and the time (s) was recorded [22];Time-orientation test: as a test that constitutes the MMSE, it evaluates the participants’ temporal orientation by asking five questions; “What year is it?”, “What month is it?”, “What date is it?”, “What day of the week is it?”, “What season is it?”;Three-item recall test: is a part of the MMSE assessing the participant’s delayed memory. Instruct the participants to listen and remember three words; tree, hat, and car, and then repeat them. After minutes with other questions as the recall distracter, ask the patient to repeat the three previously stated words [23]. Corrected answers are the score with a maximum of 3.

#### 2.4.2. The Frontal Assessment Battery (FAB)

FAB consists of 6 subtests (similarities test, verbal fluency test, Luria motor sequences, conflicting instructions, go-no go test, and prehension behavior) examining different functions related to the frontal lobes. Each subtest is scored from 0 to 3 (better score), for a maximum score of 18 [24].

### 2.5. Statistical Analysis

We used the independent t-test for continuous variables and the chi-squared test for categorical variables. The results are presented as mean ± standard deviation (SD) or as numbers (percentage, %). Sensitivity, specificity, and positive and negative likelihood ratios (with 95% confidence intervals (CIs)) were calculated for each orientation of time. Examination of the receiver operating characteristic curves was used to select the cut-off points in error scores for each aspect of three-item recalls in the supplement that maximized the sum of sensitivity and specificity.

Statistical analysis was performed using IBM SPSS Statistics Version 23.0 for Windows (IBM Corp., Armonk, NY, USA), and significance was defined as *p* < 0.05.

## 3. Results

### 3.1. Baseline Characteristics of the Study Population

Figure 1 shows the distribution according to whether the participants had MCI or depression. Participants with MCI were 592, and 575 participants had depression. A total of 172 participants had both MCI and depression, and 1673 participants had neither MCI nor depression.

Table 1 shows the baseline characteristics of participants according to the MCI and depression status. A total of 592 participants were classified into the MCI group, with a mean age of 76.6 years. In the MCI group, there were fewer men (*p* = 0.010), lower education levels (*p* < 0.001), lower urban residence (*p* < 0.001) than those in the non-MCI group. The MCI group more commonly answered the time-orientation questions incorrectly in any of the five questions. In particular, the MCI group showed the highest rate of disorientation to the year (17.7%), followed by the date (10.5%), day of the week (10.0%), season (4.1%), and month (3.5%) (Table 1).

Regarding depression, 575 participants were grouped into the depression group, with a mean age of 76.6 years. The proportion of men in the depression group was lower (32.5%) than that in the non-depression group. Errors in recognizing the year were most commonly seen (16.0%) in the depression group (Table 1).

Table 2 shows the baseline characteristics according to each time-orientation item. Age, education, and cell phone use were significantly different between the two groups. Orientations to the year (*p* < 0.001), date (*p* < 0.001), and day of the week (*p* < 0.05) were different by sex. Female were wrong more on year (male, 4.3%; female, 12.5%) and date (male, 3.4%; female, 6.9%), and male did more on day of the week (male, 7.1%; female, 4.4%). There was no significant difference between genders on month and season (Table 2).

Baseline characteristics of participants according to the MCI, depression, and each time-orientation item were analyzed separately for males and females (Appendix A).

### 3.2. Time-Orientation Tests for Diagnosing MCI

Each of the five subscales of time orientation showed similar positive predictive values (PPVs) (38.8%–46.2%), accuracies (76.5%–77.7%), and negative predictive values (NPVs) (78.2%–80.0%) for the diagnosis of MCI (Table 3). The sensitivities of the five subscales were relatively low, and disorientation to year showed the highest sensitivity (17.7%) for the diagnosis of MCI. In contrast, the three-item recall test with a cut-off of two errors, commonly regarded as a suitable screening tool for cognitive function, showed higher sensitivity (37.8%) and a similar NPV (80.9%), but a lower PPV (30.2%) and accuracy (66.8%) for the diagnosis of MCI compared to those for disorientation to the year (Appendix A).

### 3.3. Time-Orientation Tests for Diagnosing Depression

Among the time-orientation tests, “year” disorientation had the highest sensitivity (16.0%) and NPV (80.2%), but disorientation to the month showed the highest specificity (98.7%), PPV (40.4%), and accuracy (78.1%) (Table 4). In contrast, the three-item recall test with a cut-off of two errors showed higher sensitivity (32.5%) and a similar NPV (79.9%), but a lower PPV (25.2%) and accuracy (64.7%) for the diagnosis of depression compared to those for the disorientation to the year (Appendix A).

### 3.4. Time-Orientation Tests for Diagnosing MCI or Depression

For the diagnosis of MCI or depression, disorientation to the year had the highest sensitivity (15.5%), PPV (67.0%), NPV (65.5%), and accuracy (65.6%) (Table 5). In contrast, the three-item recall test with a cut-off of two errors had higher sensitivity (34.6%) and a similar NPV (66.2%) but a lower PPV (46.4%) and accuracy (60.7%) for the diagnosis of MCI or depression (Appendix A).

### 3.5. Time-Orientation Tests for Diagnosing MCI, Depression, and MCI or Depression

Each result of Table 3, Table 4 and Table 5 was analyzed separately according to sex (Appendix A). The sensitivity of year orientation was high in female (MCI, 9.9% (men) vs. 23.6% (women); depression, 9.1% (men) vs. 19.3% (women); MCI or depression, 8.6% (men) vs. 19.8% (women)). The accuracy of each time orientation for diagnosing MCI, depression, and MCI or depression was relatively high in males.

## 4. Discussion

We found that 8.6% of the community-dwelling older adults in South Korea had disorientation to the year, and the older adults with disorientation to the year had a high probability of having MCI or depression, although the sensitivity is low. To the best of the authors’ knowledge, this is the first study that if disorientation to the year is found, it means a high possibility of MCI or depression in community-dwelling elderly.

In individuals with MCI, there was a noticeable decline in cognitive abilities that do not interfere with daily functioning are at increased risk of developing Alzheimer’s disease or other dementia [25]. In one meta-analysis, MCI incidence per 1000 person-years was 22.5 for ages from 75 to 79 years, up to 60.1 for ages over 85 years [25]. Meanwhile, depression is the single largest contributor to global disability and a major contributor to suicides [26]. One meta-analysis shows that the prevalence of depression among the elderly over 60 years is 34.4% in community-based settings [26]. MCI and depression in the elderly may have varied presentations and may be difficult to diagnose. Therefore, a quick assessment tool for suspecting MCI or depression in the elderly in primary care is needed.

This study analyzed the relationship between each subscale of the time-orientation tests and MCI and depression. Although a previous study found that errors in time orientation in the elderly were related to an increased risk of future dementia [10], no study has examined the value of time-orientation tests as a case finding for MCI. In addition, one study showed that temporal disorientation remained independently associated with late-life depression in a multiple regression analysis of community-dwelling individuals aged over 75 years [27], but no studies have found a relationship between orientation in time subscales and depression.

Our study showed that the error in time orientations to the year had a relatively high PPV (45.7%) for MCI, although the sensitivity (17.7%) was lower than that of the three-item recall tests. In addition, the error in year or month showed the highest PPV of approximately 40% for depression. Taken together, disorientation to the year, out of the five time subscales, had the highest PPV (67.0%), NPV (65.5%), accuracy (65.6%), and sensitivity (15.5%) for MCI or depression diagnosis. This tendency is consistent when analyzing by sex (male, PPV (61.1%), accuracy (70.7%); female, PPV (68.8%), accuracy (61.1%)) (Appendix A and Appendix A). Therefore, if an older adult living in the community provides a wrong answer when asked the current year, they have a 45.7% probability of having MCI and 67% probability of having MCI or depression. This result is also in line with findings from other studies; errors in naming the year and month had the highest sensitivity (95%) and specificity (86.5%) for detecting cognitive impairment in older hospitalized patients [28].

It is interesting that disorientation to the year had higher accuracy and PPV for MCI or depression prediction than the three-item recall test. This is plausible because both amnestic and non-amnestic MCIs are included in the diagnosis of MCI in this study according to the definition of the National Institute on Aging and the Alzheimer’s Association.

In this study, the error rate in naming the year was very high, at 8.6%, while that for the month, date, day of the week, and the season was 1.8%, 5.2%, 5.7%, and 1.9%, respectively. Among a sample of non-institutionalized subjects aged ≥ 65 years in three cities in France, the most common error was observed for the date (5%), followed by the season (3%) and day of the week (2%); less than 1% of the participants failed to correctly name the current month or year [10]. The reason why the error rate for naming the year is so high among this cohort is not certain. Some older adults use the lunar calendar instead of the solar calendar; however, the investigations were performed from April to November, and the use of the lunar calendar may not explain this error rate. It is assumed that the error rate for naming the year is high in this cohort because elderly people usually do not have much social work and have a relatively monotonous daily life, which might mean that the changes in a year do not matter much to them. Also, further studies would be needed for a generalization of the results because a dual or triple-year system is also in use in various countries such as Israel, China, and Saudi Arabia.

In sex subgroup analysis, the female group shows relatively higher sensitivity of year orientation for diagnosing MCI, depression, and MCI or depression (Appendix A) compared to male participants. This may be because female older adults participated in social activities less than male older adults. Further studies would be needed to explain this phenomenon.

This study has some limitations. First, the age of the participants in the study was from 70 to 84 years, which might limit the applicability of our results to a general older adult population. Second, depression was defined only using the GDS score. In addition, we did not subdivide MCI into amnestic MCI and non-amnestic MCI for separate analysis. In addition, as one meta-analysis found that MoCA meets the criteria for screening tests for the detection of MCI in patients over 60 years better than MMSE [29], a future study would be needed to proceed with other screening tools for evaluating cognitive function.

The five subscales of time orientation had relatively low sensitivity for MCI screening. Nevertheless, our study showed for the first time that temporal disorientation, especially disorientation to the year, is associated with higher accuracy and PPV for MCI or depression among community-dwelling older adults. This could mean that by performing a quick assessment of orientation to the “year” in older adults who do not have dementia, clinicians can suspect the possibility of MCI or depression and then lead to further screening tests. Considering MMSE takes about 6 to 10 min to perform [30], evaluating patients’ year orientation first, which can be confirmed in about 10 s is thought to be helpful as an aid to case finding in the busy primary care practice.

## 5. Conclusions

Older adults with temporal disorientation, especially to the year, have a high probability of having MCI or depression, although the sensitivity is low. Therefore, the question “what year is it?” can be helpful as an aid to case finding to MCI or depression in community or primary care settings.

## Figures and Tables

**Figure 1 ijerph-18-08096-f001:**
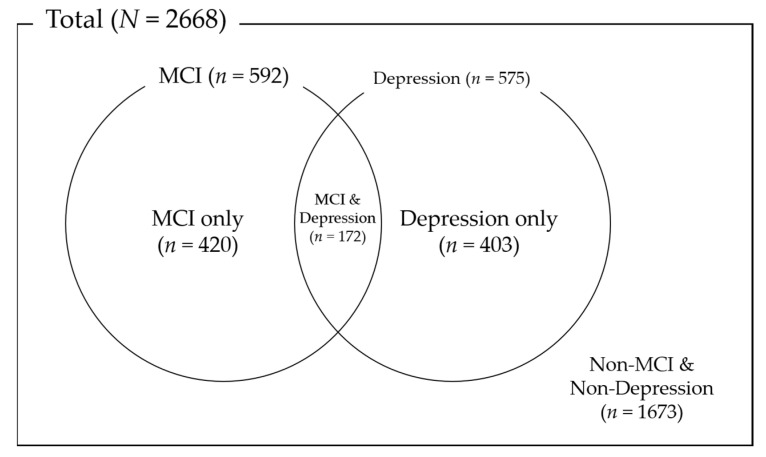
Distribution of MCI and depression.

**Table 1 ijerph-18-08096-t001:** Baseline characteristics of the study subjects according to mild cognitive impairment or depression.

Variable	Mild Cognitive Impairment		*p*	Depression		*p*	Total (*n* = 2668)
	Non-MCI(*n* = 2076)	MCI (*n* = 592)		Non-Depression (*n* = 2093)	Depression (*n* = 575)		
Age, years	75.8 ± 3.9	76.6 ± 3.8	<0.001	75.8 ± 3.9	76.6 ± 3.9	<0.001	76.0 ± 3.9
Sex							
Male	1012 (48.7)	253(42.7)	0.010	1078 (51.5)	187 (32.5)	<0.001	1265 (47.4)
Female	1064 (51.3)	339 (57.3)	0.010	1015 (48.5)	388 (67.5)	<0.001	1403 (52.6)
BMI	24.49 ± 3.02	24.37 ± 3.02	0.373	24.48 ± 3.00	24.42 ± 3.11	0.667	24.47 ± 3.02
Polypharmacy	639 (30.8)	209 (35.3)	0.037	1463 (69.9)	357 (62.1)	<0.001	848 (31.8)
Smoking	815 (39.3)	208 (35.1)	0.069	852 (40.7)	171 (29.7)	<0.001	1023 (38.3)
Alcohol drinking	390 (18.8)	88 (14.9)	0.028	406 (19.4)	72 (12.5)	<0.001	478 (17.9)
Education, ≥7 years	1251 (60.3)	279 (47.1)	<0.001	1305 (62.4)	225 (39.1)	<0.001	1530 (57.3)
Cell phone use	938 (45.2)	361 (61.0)	<0.001	917 (43.8)	382 (66.4)	<0.001	1299 (48.7)
Living alone	449 (21.6)	183 (30.9)	<0.001	417 (19.9)	215 (37.4)	<0.001	632 (23.7)
Urban residence	1537 (74.4)	370 (62.9)	<0.001	1506 (72.4)	401 (70.0)	0.254	1907 (71.9)
Medical aid	102 (5.0)	28 (4.9)	0.928	79 (3.8)	51 (9.1)	<0.001	130 (5.0)
Hypertension	1178 (56.7)	351 (59.3)	0.269	1193 (57.0)	336 (58.4)	0.538	1529 (57.3)
Dyslipidemia	719(34.6)	159 (26.9)	<0.001	684 (32.7)	194 (33.7)	0.632	878 (32.9)
Angina	131 (6.3)	32 (5.4)	0.417	127 (6.1)	36 (6.3)	0.864	163 (6.1)
Osteoarthritis	511 (24.6)	167 (28.2)	0.076	489 (23.4)	189 (32.9)	<0.001	678 (25.4)
Diabetes mellitus	447 (21.5)	132 (22.3)	0.690	458 (21.9)	121 (21.0)	0.666	579 (21.7)
Kidney disease	35 (1.7)	4 (0.7)	0.071	27 (1.3)	12 (2.1)	0.158	39 (1.5)
Time orientation							
Year, wrong	125 (6.0)	105 (17.7)	<0.001	138 (6.6)	92 (16.0)	<0.001	230 (8.6)
Month, wrong	26 (1.3)	21 (3.5)	<0.001	28 (1.3)	19 (3.3)	0.001	47 (1.8)
Date, wrong	78 (3.8)	62 (10.5)	<0.001	94 (4.5)	46 (8.0)	0.001	140 (5.2)
Day of the week, wrong	93 (4.5)	59 (10.0)	<0.001	121 (5.8)	31 (5.4)	0.721	152 (5.7)
Season, wrong	28 (1.3)	24 (4.1)	<0.001	39 (1.9)	13 (2.3)	0.541	52 (1.9)
MMSE, score	26.2 ± 2.8	24.0 ± 3.7	<0.001	26.0 ± 3.0	24.6 ± 3.5	<0.001	25.7 ± 3.2
TMT, s	69.4 ± 40.8	122.5 ± 89.7	<0.001	74.5 ± 54.1	105.6 ± 71.6	<0.001	81.2 ± 59.7
Digit span backward, score	3.6 ± 0.9	2.5 ± 1.1	<0.001	3.5 ± 1.1	3.0 ± 1.1	<0.001	3.4 ± 1.1
FAB, score	14.3 ± 2.4	11.2 ± 3.2	<0.001	13.9 ± 2.8	12.4 ± 3.0	<0.001	13.6 ± 2.9
Word list recall, score	5.9 ± 1.8	4.4 ± 2.4	<0.001	5.7 ± 2.0	5.1 ± 2.1	<0.001	5.6 ± 2.1

All values are presented as mean ± standard deviation or number (%). Depression was defined as a GDS score of ≥6. Polypharmacy was defined as taking five or more prescribed medications. Alcohol consumption was defined as ≥2 or 3 or more alcoholic drinks per week. Smoking was defined as lifetime consumption of ≥5 packs of cigarettes. Education was defined as a lifetime education period of ≥7 years. MMSE, Mini-Mental State Examination; TMT, trail-making test (out of 360 s); digit span backward (total score of 8); FAB, frontal assessment battery (total score of 18); recall test (total score of 10); GDS, Geriatric Depression Scale (range 0 to 15, higher scores represent more severe depression).

**Table 2 ijerph-18-08096-t002:** Baseline characteristics of study participants according to each time orientation.

Variables	Year		Month		Date		Day of the Week		Season	
	Wrong (*n* = 230)	Right (*n* = 2438)	Wrong (*n* = 47)	Right (*n* = 2621)	Wrong (*n* = 140)	Right (*n* = 2528)	Wrong (*n* = 152)	Right (*n* = 2516)	Wrong (*n* = 52)	Right (*n* = 2616)
Age, years	77.4 ± 4.0	75.8 ± 3.8 †	78.5 ± 4.0	76.0 ± 3.9 †	78.0 ± 3.8	75.8 ± 3.9 †	77.4 ± 4.2	75.9 ± 3.9 †	77.7 ± 4.0	75.9 ± 3.9 *
Sex										
Male	54 (4.3)	1211 (95.7) †	16 (1.3)	1249 (98.7)	43 (3.4)	1222 (96.6) †	90 (7.1)	1175 (92.9) *	26 (2.1)	1239 (97.9)
Female	176 (12.5)	1227 (87.5) †	31 (2.2)	1372 (97.8)	97(6.9)	1306 (93.1) †	62 (4.4)	1341 (95.6) *	26 (1.9)	1377 (98.1)
BMI	24.4 ± 3.2	24.5 ± 3.0	24.3 ± 3.3	24.5 ± 3.0	24.3 ± 3.3	24.5 ± 3.0	24.0 ± 3.2	24.5 ± 3.0	24.2 ± 3.8	24.5 ± 3.0
Polypharmacy	91 (39.6)	757 (31.1) *	18 (38.3)	830 (31.7)	49 (35.0)	799 (31.6)	55 (36.2)	793 (31.5)	15 (28.8)	833 (31.8)
Smoking	57 (24.8)	966 (39.6) †	13 (27.7)	1010 (38.5)	38 (27.1)	985 (39.0) *	73 (48.0)	950 (37.8) *	16 (30.8)	1007 (38.5)
Alcohol drinking	26 (11.3)	452 (18.5) *	5 (10.6)	473 (18.0)	19 (13.6)	459 (18.2)	23 (15.1)	455 (18.1)	14 (26.9)	464 (17.7)
Education, ≥7 years	48 (20.9)	1482 (60.8) †	13 (27.7)	1517 (57.9) †	48 (34.3)	1482 (58.6) †	67 (44.1)	1463 (58.1) *	22 (42.3)	1508 (57.6) *
Cell phone use	168 (73.0)	1131 (46.4) †	34 (72.3)	1265 (48.3) †	96 (68.6)	1203 (47.6) †	93 (61.2)	1206 (47.9) †	32 (61.5)	1267 (48.4) †
Living alone	89 (38.7)	543 (22.3) †	13 (27.7)	619 (23.6)	53 (37.9)	579 (22.9) †	37 (24.3)	595 (23.6)	11 (21.2)	621 (23.7)
Urban residence	132 (57.4)	1775 (73.3) †	30 (63.8)	1877 (72.0)	80 (57.1)	1827 (72.7) †	93 (62.0)	1814 (72.5) *	28 (54.9)	1879 (72.2) *
Medical aid	12 (5.4)	118 (4.9)	2 (4.7)	128 (5.0)	10 (7.4)	120 (4.8)	11 (7.3)	119 (4.8)	0 (0.0)	130 (5.0)
Hypertension	145 (63.0)	1384 (56.8)	28 (59.6)	1501 (57.3)	80 (57.1)	1449 (57.3)	91 (59.9)	1438 (57.2)	28 (53.8)	1501 (57.4)
Dyslipidemia	67 (29.1)	811 (33.3)	11 (23.4)	867 (33.1)	48 (34.3)	830 (32.8)	40 (26.3)	838 (33.3)	16 (30.8)	862 (33.0)
Angina	16 (7.0)	147 (6.0)	0 (0.0)	163 (6.2)	8 (5.7)	155 (6.1)	13 (8.6)	150 (6.0)	2 (3.8)	161 (6.2)
Osteoarthritis	78 (33.9)	600 (24.6) *	16 (34.0)	662 (25.3)	43 (30.7)	635 (25.1)	41 (27.0)	637 (25.3)	14 (26.9)	664 (25.4)
Diabetes mellitus	42 (18.3)	537 (22.0)	11 (23.4)	568 (21.7)	38 (27.1)	541 (21.4)	34 (22.4)	545 (21.7)	14 (26.9)	565 (21.6)
Kidney disease	3 (1.3)	36 (1.5)	0 (0.0)	39 (1.5)	0 (0.0)	39 (1.5)	2 (1.3)	37 (1.5)	0 (0.0)	39 (1.5)
MCI	105 (45.7)	487 (20.0) †	21 (44.7)	571 (21.8) †	62 (44.3)	530 (21.0) †	59 (38.8)	533 (21.2) †	24 (46.2)	568 (21.7) †
MMSE, score	20.8 ± 4.0	26.2 ± 2.7 †	18.8 ± 4.8	25.9 ± 3.0 †	21.5 ± 4.2	26.0 ± 2.9 †	22.3 ± 4.3	25.9 ± 3.0 †	20.6 ± 5.5	25.8 ± 3.0 †
TMT, s	153.5 ± 94.7	74.4 ± 50.1 †	147.5 ± 104.2	80.0 ± 57.9 †	136.0 ± 91.6	78.2 ± 55.9 †	115.2 ± 84.4	79.1 ± 57.2 †	124.6 ± 86.2	80.3 ± 58.7 *
Digit span backward, score	2.4 ± 1.3	3.4 ± 1.0 †	2.5 ± 1.4	3.4 ± 1.1 †	2.7 ± 1.3	3.4 ± 1.1 †	3.0 ± 1.3	3.4 ± 1.1 *	2.9 ± 1.7	3.4 ± 1.1 *
FAB, score	10.9 ± 3.0	13.9 ± 2.7 †	10.9 ± 3.1	13.6 ± 2.9 †	11.3 ± 3.1	13.7 ± 2.8 †	11.9 ± 3.2	13.7 ± 2.8 †	11.3 ± 3.9	13.6 ± 2.8 †
Word list recall, score	3.7 ± 2.0	5.8 ± 2.0 †	3.5 ± 2.2	5.6 ± 2.0 †	3.8 ± 2.2	5.7 ± 2.0 †	4.0 ± 2.2	5.7 ± 2.0 †	4.4 ± 2.2	5.6 ± 2.1 †

Table 2 shows the comparison between participants who answered right and wrong for each five time-orientation domains. All values are presented as mean ± standard deviation or number (%). Depression was defined as a GDS score of ≥ 6. Polypharmacy was defined as taking five or more prescribed medications. Alcohol consumption was defined as ≥2 or 3 or more alcoholic drinks per week. Smoking was defined as lifetime consumption of ≥5 packs of cigarettes. Education was defined as a lifetime education period of ≥7 years. MMSE, Mini-Mental State Examination; TMT, trail-making test (out of 360 s); digit span backward (total score of 8); FAB, frontal assessment battery (total score of 18); recall test (total score of 10); GDS, Geriatric Depression Scale (range 0 to 15, higher scores represent more severe depression). * *p* < 0.05; † *p* < 0.001.

**Table 3 ijerph-18-08096-t003:** Time-orientation tests for the diagnosis of MCI.

Time Orientation	Sensitivity	Specificity	PPV	NPV	Accuracy
Year (wrong)	17.7%	94.0%	45.7%	80.0%	77.1%
Month (wrong)	3.5%	98.7%	44.7%	78.2%	77.6%
Date (wrong)	10.5%	96.2%	44.3%	79.0%	77.2%
Day of the week (wrong)	10.0%	95.5%	38.8%	78.8%	76.5%
Season (wrong)	4.1%	98.7%	46.2%	78.3%	77.7%

PPV, positive predictive value; NPV, negative predictive value; Accuracy, proportion of true results among the total number of cases examined.

**Table 4 ijerph-18-08096-t004:** Time-orientation tests for the diagnosis of depression (GDS score of ≥6).

Time Orientation	Sensitivity	Specificity	PPV	NPV	Accuracy
Year (wrong)	16.0%	93.4%	40.0%	80.2%	76.7%
Month (wrong)	3.3%	98.7%	40.4%	78.8%	78.1%
Date (wrong)	8.0%	95.5%	32.9%	79.1%	76.6%
Day of the week (wrong)	5.4%	94.2%	20.4%	78.4%	75.1%
Season (wrong)	2.3%	98.1%	25.0%	78.5%	77.5%

GDS, Geriatric Depression Scale (range 0 to 15, higher scores represent more severe depression).

**Table 5 ijerph-18-08096-t005:** Time-orientation tests for the diagnosis of MCI or depression.

Time Orientation	Sensitivity	Specificity	PPV	NPV	Accuracy
Year (wrong)	15.5%	95.5%	67.0%	65.5%	65.6%
Month (wrong)	3.0%	99.0%	63.8%	63.2%	63.2%
Date (wrong)	8.7%	96.8%	62.1%	64.1%	64.0%
Day of the week (wrong)	7.8%	95.6%	51.3%	63.6%	62.9%
Season (wrong)	3.1%	98.7%	59.6%	63.2%	63.1%

## Data Availability

The data presented in this study are available on request from the corresponding author. The data are not publicly available due to privacy reasons.

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
