# Peer review of "Usefulness of Orientation to the Year as an Aid to Case Finding of Mild Cognitive Impairment or Depression in Community-Dwelling Older Adults"

_ijerph, 2021, doi:10.3390/ijerph18158096_

Round 1

Reviewer 1 Report

I have no further comments. 

Author Response

Our manuscript could be progressed in a better way thanks to you. Thank you for the  reviewer's considerable opinion. 

Reviewer 2 Report

It is worth that the authors taken into account in their future research this more sensitivity and specificity method in cognitive function evaluation as Montreal Cognitive Assessment (MoCA) .

Author Response

Point 1: It is worth that the authors taken into account in their future research this more sensitivity and specificity method in cognitive function evaluation as Montreal Cognitive Assessment (MoCA) .

Response 1: We would be grateful to have the opportunity to proceed the study with MoCA test next time. 

We have added this in the limitation of discussion part (L289-291).

"And also, as one meta-analysis found that MoCA meets the criteria for screening tests for the detection of MCI in patients over 60 years better than MMSE [29], future study would be needed to proceed with other screening tools for evaluating cognitive function."

Thank you for the considerable suggestion.

Reviewer 3 Report

All of my comments were taken into consideration by the authors.

Author Response

Thanks to your thoughtful advice, our manuscript has been completed for the better. Thank you for the considerable suggestion. 

This manuscript is a resubmission of an earlier submission. The following is a list of the peer review reports and author responses from that submission.

Round 1

Reviewer 1 Report

In this interesting manuscript the authors investigate in a reasonably large cohort the relationship between year orientation and MCI and dementia. They found that error in year orientation is indicative of decline of MCI and of depression.

Strengths of the manuscript

The authors use “mostly” a clear language to describe the cohort and the investigation.

Weakness points

Major points

  1. Perhaps one of the limitations of this study, is that in Korea, the Korean traditional calendar (시헌력 (時憲暦) is still maintained. In that case it could be expected from older individuals to use the year system that they are usually familiar with. Thus not knowing the current year according to the questionnaire preference might not be indicative of cognitive decline.

  1. Furthermore, a dual or triple-year system is also in use in various countries such as Israel, China, and Saudi Arabia. Thus a generalization of the authors' findings would need to be further validated.

  1. Another major limitation is the originality of the research, time orientation has been already discussed as a major indicator of dementia (see for example J Alzheimers Dis. 2016 Jul 1;53(4):1411-8. doi: 10.3233/JAD-160295. Time Orientation and 10 Years Risk of Dementia in Elderly Adults: The Three-City Study). Thus the main focus of the manuscript is not clear.

Minor points

  1. a) Methods
  2. It is not clear if there is a difference between males and females regarding the year orientation question
  3. The control group's total number is not clear.

3, The data, authors have is large, thus more types of tests could be done (stratification based on gender, focus more on depression relationship to MCI, and so on)

  1. If the main focus of the manuscript is the time orientation, depression and MCI, then it would be interesting what is the relationship between these three variables eg. Using ANOVA .
  2. b) Results

The results only cover time orientation but the results of the other tests presented in the methods section are not highlighted.

  1. c) Discussion

The discussion section lacks a section explaining the significance of the results and how they could be applied in a larger context.

Reviewer 2 Report

Mild cognitive impairment (MCI) by many researchers consider as a transitional stage between the natural aging and dementia. Early identification and intervention for MCI may help to slow down the development of dementia. Screening is akey step in the diagnosis of dementia, which is why methods used in them should have a high sensitivity for the detection of MCI. In this study, the neuropsychological tests conducted by the authors characterized by low sensitivity and thus limited functionality in recognition of MCI or depression.

Although, the Mini-Mental State Examination (MMSE) is the most commonly used scale in cognitive function evaluation, it is claimed to be imprecise for MCI detection. Recent studies show that Montreal Cognitive Assessment (MoCA) test better meets the criteria for screening tests for the detection of MCI among patients over 60 years of age than MMSE. It is worth that the authors  taken into account in their future research this more sensitivity and specificity method in cognitive function evaluation.

Comments and Suggestions

Abstract

The abstract should state the main problem, methods, results, and conclusions. The abstract must be factual and comprehensive.

 Introduction

Line 37-40 “In addition, the dementia prodrome, mild cognitive impairment (MCI), has garnered attention and active study since the early 1990s as a risk factor for developing dementia, with an annual rate of progression of 12%, reaching up to 20% in populations at higher risk [4, 5]. Who is in the higher risk population?

Materials and Methods

- In Line 70 it was reported that 3014 adults were recruited, and below line 74 there is information that 1500 women and the same number of men. What happened with 14 people ?

- What was total number of healthy people in the control group?

- A better name for 2.2. section would be „Neuropsychological tests or Neuropsychological screening scales” than "Measures" and sequentially described the tests i.e.

2.2.1 Alzheimer's Disease  Assessment Battery (CERAD-K) 

2.2.2 Frontal Assessment  Battery (FAB)  …

 Results

Very modest analysis of the results despite quite a large amount of data collection. The results was described generally and a lot of important data was omitted.

Why were the tests results presented only from a men group ?

The results section should be correlate with the Materials and Method section. This is missing here!

Line 155-156 “Regarding  depression,  participants  were  grouped  into  the  depression  group, with a mean age of 76.6 years. The proportion of men in the depression group was lower (32.5%) than that in the non-depression group”. Who was in non-depression group? healthy people? Please, clarify this.

Line 172 - What was compared with what in Table 2 ? (p<0.05; †p<0.001)

The tables are numbered 1-5, which means Table S2 or S3? Where should the reader look for them?

This part of the manuscript should be improved.

Discussion

 The weakest part of the manuscript is discussion

Authors should discuss the results in the broadest context possible and relate them to the other, previous studies and of the working hypotheses. It is  worth mentioning here the future research directions. The discussion should be rewritten.

Reviewer 3 Report

First of all, congratulations to authors for seeking practical and easy ways to make prevention more effective and available in the context of real life demanding. All in, the paper is valuable. Please find some suggestions from my part:

  1. L57 “Primary care physicians have detected only 40–50% of patients with depression among older adults” awkwardness of style, it is generalized data, not a specific number, please consider changing into more academic language.
  2. First citation [1] may be seen as indirect quotation which should be avoided, consider changing for the original source.
  3. Putting both [18] and [20] references is not justified, what does it bring? All the information needed about CERAD-K could be taken from one of those. Suggestion to avoid putting two sources for such a technical matter. Same applies to [18] with [21] and [18] with [22]. Could you shed some light why you decided to do so? Otherwise please change it.
  4. In the discussion, I suggest authors have more discussions about the findings of study and future work.
  5. I miss more information about what demographic data were gathered.